# Evaluating the feasibility of 12-lead electrocardiogram reconstruction from limited leads using deep learning

Oriana Presacan [1,10], Alexandru Dorobanțiu [2,10], Jonas L. Isaksen [3], Tobias Willi[4], Claus Graff[5], Michael A. Riegler [6], Arun R. Sridhar [7,10], Jørgen K. Kanters [3,8,10] & Vajira Thambawita [9,10] ✉

## Abstract

**Background** Wearables with integrated electrocardiogram (ECG) acquisition have made single-lead ECGs widely accessible to patients and consumers. However, the 12-lead ECG remains the gold standard for most clinical cardiac assessments. In this study, we developed a neural network to reconstruct 12-lead ECGs from single-lead and dual-lead ECGs, and evaluated the mathematical accuracy.

**Methods** We used lead I or leads I and II from 9514 individuals from the Physikalisch-Technische Bundesanstalt (PTB-XL) cohort and a generative adversarial network, with the aim of recreating the missing leads from the 12-lead ECG. ECGs were divided into training, validation, and testing (10%). Original and recreated leads were measured with a commercially available algorithm. Differences in means and variances were assessed with Student's t-tests and F-tests, respectively. Calibration and bias were assessed with Bland-Altman plots. Inter-lead correlations were compared in original and recreated ECGs.

**Results** The variability of precordial ECG amplitudes is significantly reduced in recreated ECGs compared to real ECGs (all $p < 0.05$), indicating regression-to-the-mean. Amplitude averages are recreated with bias ($p < 0.05$ for most leads). Reconstruction errors depend on the real amplitudes, suggesting regression-to-the-mean ($R^2$ between target and error in R-peak amplitude in lead V3: 0.92). The relations between lead markers have a similar slope but are much stronger due to reduced variance (R-peak amplitude $R^2$ between leads I and V3, real ECGs: 0.04, recreated ECGs: 0.49). Using two leads does not significantly improve 12-lead recreation.

**Conclusions** AI-based 12-lead ECG reconstruction results in a regression-to-the-mean effect rather than personalized output, rendering it unsuitable for clinical use.

## Plain language summary

Electrocardiogram (ECG) testing measures electrical signals from the heart and can be used to diagnose and monitor people with heart problems. Wearable devices have made ECG testing widely available for patients. However, ECG tests using only a single-lead to measure electrical signals from the heart lack the clinical detail provided by a 12-lead ECG test. In our study, we explored the use of artificial intelligence (AI) to convert single- or dual-lead ECGs into 12-lead ECGs and assessed the accuracy of the reconstructed ECGs using mathematical metrics and an AI algorithm. While the reconstructed ECGs appeared visually normal, we found that the AI algorithm generated the missing leads based on population averages rather than individual patient characteristics. These results reveal that lead conversion using AI is not reliable and should not be used clinically.

Electrocardiography is a simple, low-cost, and quick method for assessing the electrical activity of the heart in order to obtain clinical information about heart diseases. Over a span of ten seconds, this method captures the heart's voltage using ten electrodes. Given the inherently three-dimensional (3D) composition of the human body, multiple electrodes are required to record the 3D electrical activity of the heart. From a mathematical perspective, only three orthogonal leads should be required to characterize the electrical

dipolar vector in space; a concept well-known from the 3-lead Frank vectorcardiogram[1].

The Frank vectorcardiogram simplifies the complexity of the heart's electrical activity by treating it as an infinitesimal small stationary dipole in a large, uniform conducting medium. While this is a simplification, it allows for the creation of a model to understand the heart's electrical activity. The standard clinical electrocardiogram (ECG) consists of 12 leads, with six leads originating from the extremities (four of these being mathematically

[1]Oslo Metropolitan University, 0167 Oslo, Norway. [2]Lucian Blaga University of Sibiu, 550024 Sibiu, Romania. [3]University of Copenhagen, 2200 Copenhagen N, Denmark. [4]KTH Royal Institute of Technology, 11428 Stockholm, Sweden. [5]Aalborg University, 9220 Aalborg Ø, Denmark. [6]Simula Research Laboratory, Kristian Augusts gate 23, 0164 Oslo, Norway. [7]Pulse Heart Institute, Multicare Health System, Tacoma, WA, USA. [8]University of California, San Francisco, USA. [9]SimulaMet, Stensberggata 27, 0170 Oslo, Norway. [10]These authors contributed equally: Oriana Presacan, Alexandru Dorobanțiu, Arun R. Sridhar, Jørgen K. Kanters, Vajira Thambawita. ✉e-mail: vajira@simula.no

(a) Generator

(b) Discriminator

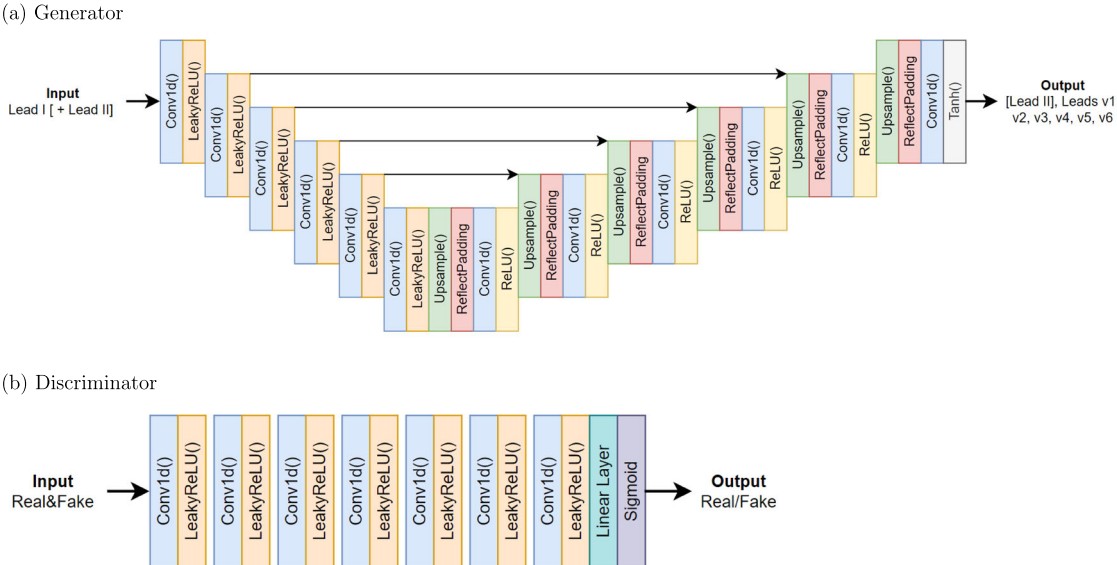

**Fig. 1 | GAN architecture for reconstructing ECGs using Lead I or Leads I and II.**
**a** The generator takes Lead I or Leads I and II as input and reconstructs the additional electrocardiogram (ECG) leads V1, V2, V3, V4, V5, and V6 as output. **b** The discriminator receives real and synthesized (fake) ECGs as input and classifies them as real or fake. Each block represents a layer type, with colors distinguishing them. The generator follows an encoder-decoder structure, using one-dimensional convolutional layers (Conv1d()), Leaky Rectified Linear Unit activation (LeakyReLU()) and Rectified Linear Unit activation (ReLU()), reflection padding (ReflectPadding), and up-sampling layers (Upsample()), ending with a hyperbolic tangent activation (Tanh()). The discriminator consists of stacked one-dimensional convolutional layers (Conv1d()) with Leaky Rectified Linear Unit activation (LeakyReLU()), followed by a linear layer and sigmoid activation (Sigmoid()).

redundant) and six leads from electrodes placed on the chest (precordial electrodes).

Principal Component Analysis (PCA), a statistical method that simplifies data by identifying key variables (principal components), demonstrates that 98% of the 12-lead ECG information in healthy subjects is captured by the first three principal components[2]. In patients with acute or chronic cardiac disease, the information in the first three principal components unfortunately decreases. As the heart is not an ideal dipole, because of its size and its movement during the heart cycle, the precordial leads not only have information about the Z-axis of the 3D electrical vector but also contain local electrical non-dipolar information from cardiac tissue just below the electrode. An example of important non-dipolar information is fractionated QRS complexes, which may demand local information from specific single leads and can not necessarily be constructed solely from other leads[3].

Whereas 12-lead ECGs are the most common type of ECGs, in the acute context and for long-term recordings, a reduced set of electrodes, typically consisting of one or two leads, is utilized. The reduced leads contain nearly the same information in the time dimension, making them well-suited for arrhythmia analysis. However, the lack of dimensions may result in substantial loss of spatial information, which is essential for diagnosing myocardial infarction, hypertrophy, repolarization changes, and other localized cardiac disorders.

With the advent of one-lead portable ECG devices, there has been a renewed interest in the possibility of obtaining a full 12-lead ECG from single-lead devices. Because the spatial differences of these intervals are minimal, temporal information such as heart rate and intervals may be reliably measured from a single lead. This is not the case with amplitudes, which vary considerably depending on the angle between the lead vector's direction and the 3D cardiac voltage vector. As a result, interval measurements from one-lead wearable devices are reliable, but one-lead devices cannot currently detect amplitude changes, for example, during cardiac ischemia and myocardial infarction. That raises the question of whether it is possible to use deep learning networks[4] to create reliable 12-lead ECG from a single lead, as previously suggested[5,6], to be implemented in one-lead

wearables to detect cardiac ischemia. Advances in deep learning and neural networks have particularly spurred efforts in this area and the concept of reconstruction has spawned a number of publications in the field[5–8]. Sohn et al.[9] proposed a three-lead chest device with four electrodes and a recurrent neural network employing long short-term memory (LSTM) to generate the missing leads. Both Grande et al.[10] and Hussein et al.[11] suggested using an artificial neural network (ANN) to generate the remaining leads from the initial three leads.

From a linear mathematical perspective, it is impossible to reconstruct a full 12-lead ECG from a single lead, since a lead contains no information on the two orthogonal directions required to reconstruct the 3D vector. Nevertheless, it has been claimed that 3D reconstruction from a single input using deep learning can be reliably performed. The purpose of this investigation is to evaluate the ability of neural networks to reconstruct a full 12-lead ECG from data obtained from single-lead or two-lead ECGs. Due to the advent of two-lead portable ECG devices in the consumer market, we investigated whether the reconstruction of 12-lead ECG from two-lead ECGs is better than with a single-lead.

The study's main contributions include the implementation of generative adversarial network (GAN) models to reconstruct missing ECG leads using two input configurations: Lead I alone and a combination of Leads I and II. Performance is assessed using the Physikalisch-Technische Bundesanstalt (PTB-XL) dataset, revealing that while GAN-generated ECGs appear visually normal, they exhibit significant deficiencies compared to real ECGs. Specifically, the reconstructed precordial ECGs demonstrate reduced variability, reflecting a tendency to regress toward the population mean (all $p < 0.05$). Amplitude averages are recreated with bias ($p < 0.05$ for most leads), and reconstruction errors depend on the real amplitudes, further indicating regression-to-the-mean ($R^2 = 0.92$ for R-peak amplitude errors in Lead V3). The relationships between lead markers maintain a similar slope but appear significantly stronger in the recreated ECGs due to reduced variance ($R^2$ for R-peak amplitude between Leads I and V3: 0.04 in real ECGs vs. 0.49 in recreated ECGs). These findings highlight the limitations of deep learning-based methods in capturing individual variations and clinically significant features in ECG lead reconstruction.

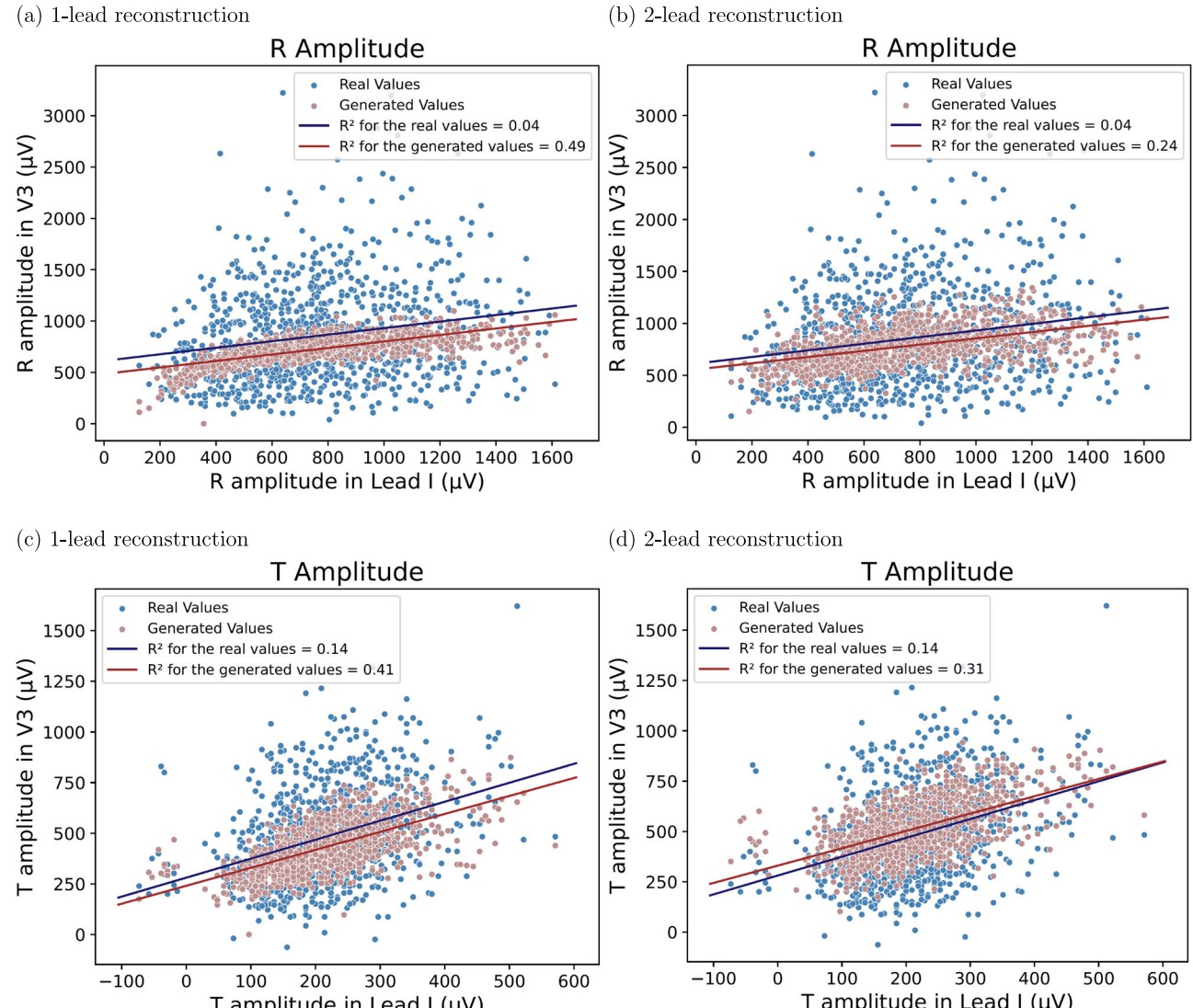

**Fig. 2 | Relationship between amplitudes in lead I and lead V3 for R-waves and T-waves for real and generated ECG data.** Relationship between Amplitudes in lead I and lead V3 for R-waves (**a**) and (**b**) and T-waves (**c**) and (**d**). Panels (**a**) and (**c**) show results from the 1-lead reconstruction, while panels (**b**) and (**d**) present results from the 2-lead reconstruction ($n$ = 952 independent samples). Each point represents an individual sample, with real values shown in blue and generated values in red. The lines indicate linear regression fits for the real (blue) and generated (red) data. The coefficient of determination ($R^2$) is reported for both cases. All amplitudes are measured in microvolts (μV).

## Methods

### Data

The models were trained and tested using the PTB-XL v1.0.3[12]. This publicly available dataset consists of anonymized 12-lead ECGs obtained from 18,869 people. The dataset was approved for open-access publication by the Institutional Ethics Committee of the Physikalisch-Technische Bundesanstalt (PTB). Furthermore, the use of anonymized data from PTB-XL was approved by the Institutional Ethics Committee, and need for informed consent from people whose data was included was waived. We obtained no further approval because we worked with retrospective, de-identified data with no possibility to impact patient care. The dataset includes ECGs from both healthy individuals and patients with various pathologies. Expert cardiologists have labeled each record, assigning different conditions to them. For our study, we specifically focused on utilizing the ECGs categorized as 'normal', which amounted to a total of 9514 patients. Their ages vary from 2 to 95 years, having a mean and standard deviation of 52.86 ± 22.25. Among them, 46% are male and 54% are female.

Each ECG record in the dataset has a duration of 10 seconds and a sampling frequency of 500 Hz. The dataset was divided into three subsets: 80% for training (7611 samples), 10% for validation (951 samples), and 10% (952 samples) for testing purposes. The performance evaluation of the models was conducted on the PTB-XL test dataset[12]. The results were assessed using the MUSE 12SL[13], which provides ECG measurements and arrhythmia diagnosis. The PTB-XL dataset contains sample values expressed in millivolts (mV). However, before passing the samples to the MUSE system, we rescaled them to microvolts (μV) since the system only accepts input in this unit. Hence, all results are expressed in microvolts (μV).

Initially, only Lead I was used as the input, and predictions were made for the remaining seven independent leads (Lead II, V1, V2, V3, V4, V5, V6). Subsequently, we aimed to assess whether employing both Lead I and Lead II as input and reconstructing only V1–V6 leads would result in better performance. Therefore, we trained additional models for this purpose.

### Models

Two UNet[14] architecture models were utilized: one employed a single lead (Lead I) as input to generate the seven missing leads (Lead II, V1–V6), while the second model utilized two leads (Leads I and II) to reconstruct the precordial leads (V1–V6). During their training, two

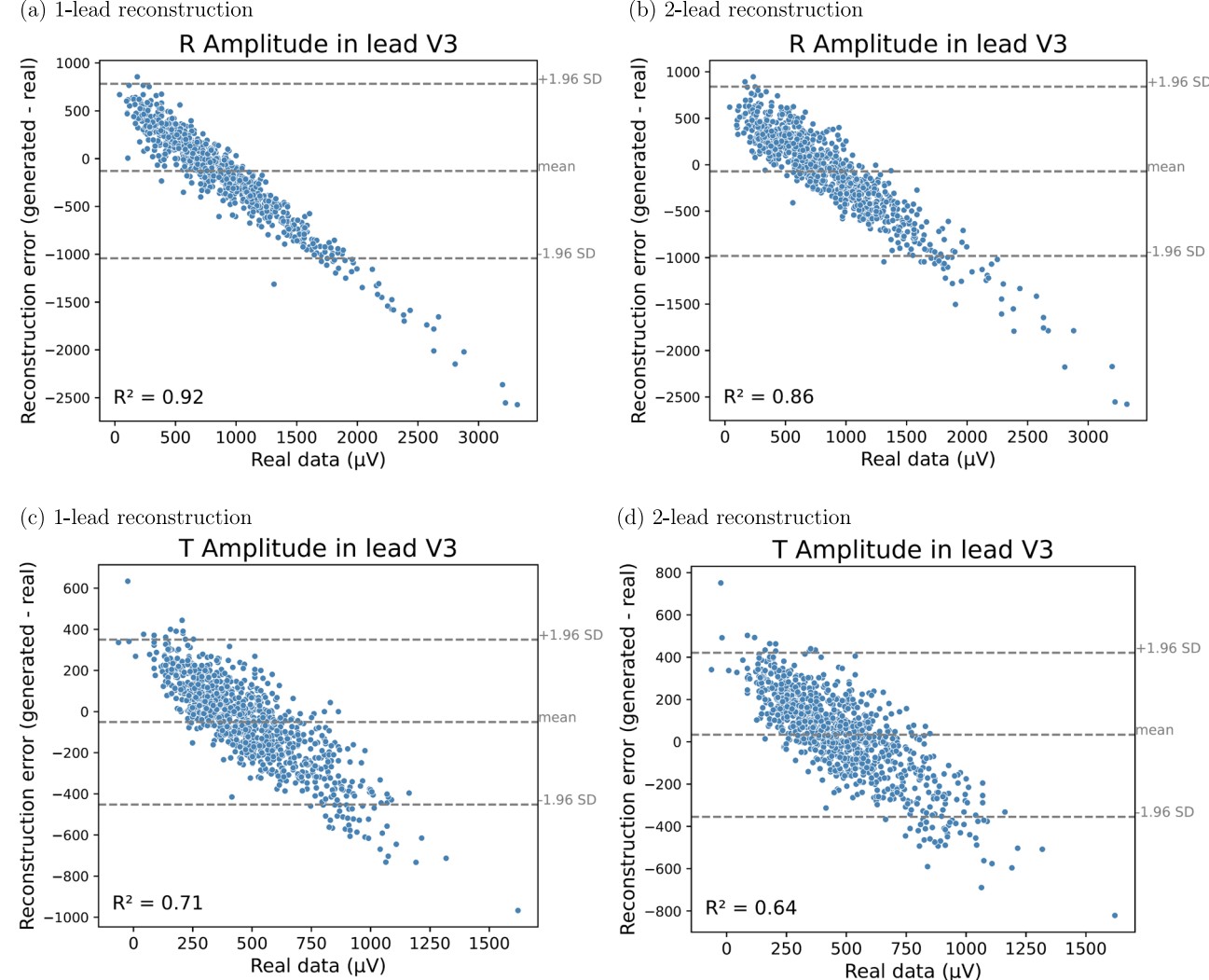

**Fig. 3 | Bland-Altman plots of the R and T amplitudes in Lead V3 for real and generated ECG data. a, b** R peak amplitude in Lead V3, **c, d** T peak amplitude in Lead V3. Panels (**a**) and (**c**) show results from the 1-lead reconstruction, panels (**b**) and (**d**) present results from the 2-lead reconstruction ($n = 952$ independent samples). The plots display the reconstruction error (generated - real) against the amplitude of the real ECGs in lead V3. Each point represents a sample. The dashed lines denote the mean error and the limits of agreement at ±1.96 standard deviations. The coefficient of determination ($R^2$) quantifies the correlation between real and generated data. Amplitudes are measured in microvolts (μV).

distinct loss functions were used: an adversarial loss derived from a discriminator model combined with an L1 loss, and a mean-squared error (MSE) for the pure U-Net model. To maintain clear differentiation between these techniques, we designate the model utilizing the adversarial loss as the GAN, and the model with MSE loss as the U-Net. Both loss techniques yielded comparable performance, with the GAN slightly outperforming the U-Net. To simplify the presentation, we only include the GAN values in the Results section. The results for the U-Net are provided in Table S1 of the Supplementary material.

GAN, initially introduced in ref. 15, consists of two neural networks—a generator and a discriminator—that are trained together in a competitive manner, where they continuously learn from each other's outputs and improve over time. The proposed generator takes as input either Lead I or Leads I and II of an ECG and produces the remaining leads. The discriminator's role is to differentiate between real and generated signals. Through training, the generator learns to produce realistic ECGs that the discriminator cannot differentiate from genuine ECGs.

**Architectures.** The models' architectures are inspired by the Pulse2Pulse model[16] that was used to generate $8 \times 5000$ ECGs from an $8 \times 5000$ random noise vector. However, we have adapted our generator (and the basic network for the UNet) to accept either a single ECG lead ($1 \times 5000$) or two ECG leads ($2 \times 5000$) signals as input and produce the corresponding missing leads. The architecture comprises a down-sampling component and an up-sampling component. The down-sampling part reduces the input signal's dimensionality while capturing the most important features, whereas the up-sampling component reconstructs the missing leads using the learned features. The former includes six 1D-convolution layers, each followed by a Leaky ReLU activation function[17], as illustrated in Fig. 1a. The latter consists of an additional six convolution layers and a ReLU activation. The arrows depicted in Fig. 1a represent the skip connections of the UNet that connect the blocks of the two components, which led to improved outcomes.

The discriminator's role is to classify the ECGs as either real or fake. To efficiently train it, we implemented a patch discriminator approach, as proposed in[18], where the discriminator evaluates patches instead of the entire image. While the Pix2Pix model applies the patch convolutionally across the entire image, our customized patch discriminator randomly selects a patch of a predetermined size (e.g., 8 channels × 800 samples) for each training batch. This strategy reduces training time and moderates the discriminator's learning rate, ensuring a balanced training process with the generator, as the discriminator initially learns faster than the generator. The

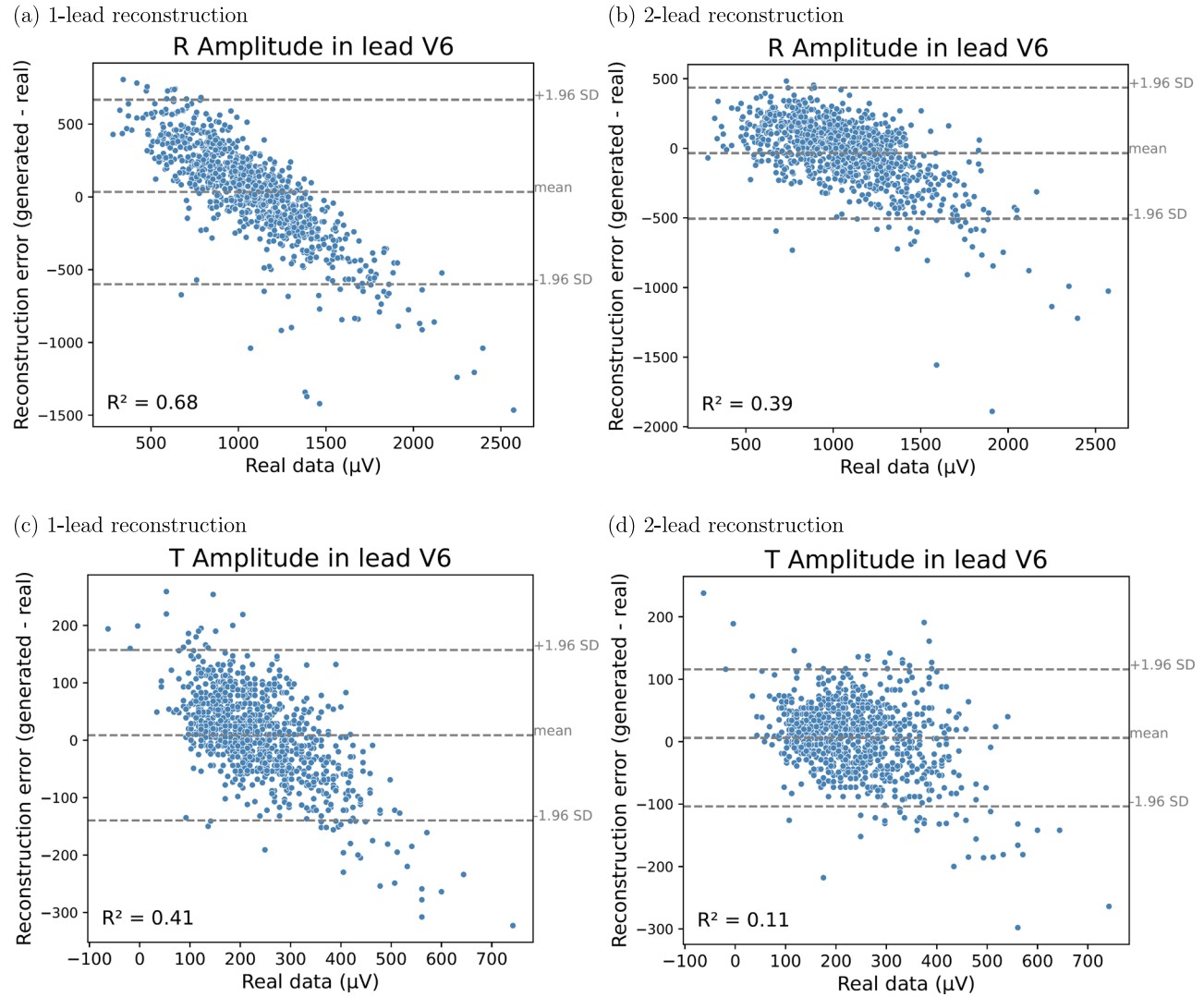

(a) 1-lead reconstruction

(b) 2-lead reconstruction

(c) 1-lead reconstruction

(d) 2-lead reconstruction

**Fig. 4 | Bland-Altman plots of the R and T amplitudes in Lead V6 for real and generated ECG data. a**, **b** R peak amplitude in Lead V6, **c**, **d** T peak amplitude in Lead V6. Panels (**a**) and (**c**) show results from the 1-lead reconstruction, panels (**b**) and (**d**) show results from the 2-lead reconstruction ($n = 952$ independent samples). The plots display the reconstruction error (generated - real) against the amplitude of the real electrocardiograms in lead V6. Each point represents a sample. The dashed lines denote the mean error and the limits of agreement at ±0.96 standard deviations. The coefficient of determination ($R^2$) quantifies the correlation between real and generated data. Amplitudes are measured in microvolts (μV).

discriminator architecture depicted in Fig. 1b comprises seven convolutional layers, each followed by a Leaky ReLU activation function.

**Training.** To train the networks a Ubuntu workstation with two Xeon processors and a GeForce NVIDIA RTX 2080ti were utilized. The PyTorch library[19] was employed for implementation. The GAN is using the Adam optimizer[20], with a learning rate of 0.0001, $\beta_1$ value of 0.5, and $\beta_2$ value of 0.9. Due to training instability, a smaller learning rate of 0.00005 was used for the U-Net. A batch size of 32 was used for both models. The U-Net employs the mean squared error (MSE) loss function. For the GAN, we utilized the WGAN-GP technique with gradient clipping[21], combined with the objective proposed in ref. 18. This training objective combines the discriminator loss with the L1 loss, which measures the error between the real and generated signals. Therefore, in addition to fooling the discriminator, the generator is also required to minimize the L1 loss. The discriminator's parameters are updated after every batch, while the generator's are updated after every other batch. To prevent overfitting, dropout was applied to three of the layers during training[22]. The networks were trained for a total of 1000 epochs. The final models were chosen based on the validation error.

**Statistics and reproducibility**

This study used statistical analyzes to evaluate the accuracy of ECG signal reconstruction, with all analyzes performed on the test dataset comprising 952 normal ECG samples from the PTB-XL dataset. All the tests are two-sided. Reproducibility was ensured through standardized data pre-processing and model training protocols, treating each ECG sample as an independent replicate ($n = 952$).

We assessed the signal average reconstruction error on a lead-by-lead basis. This was done by calculating the square root of the mean squared error (RMSE) between the real and reconstructed ECG. Based on the 12SL measurements, we analyzed the mean error and standard deviation of errors for the R peak, S peak, center ST segment (STM), and T peak amplitudes lead-by-lead. We used the subset of leads V1, V2, V3, and V6.

We utilized Student's t-test and the F-test to assess differences in means and variances, respectively. The Student's t-test was chosen as it is a robust statistical tool used to determine whether the means of two groups of data are significantly different from each other. This test is particularly suitable when dealing with normally distributed data, which is an assumption we are working with for our ECG data. The F-test was used to evaluate the equality of variances. Variance is a measure of dispersion in a dataset and can be

**Fig. 5 | Visualization of real and reconstructed ECG.** ECG plots depicting a real normal sample from the PTB-XL test set (black), alongside the reconstructed ECG (blue). The reconstructed ECG was generated using one lead (**a**) and two leads (**b**) with the generative adversarial network (GAN).

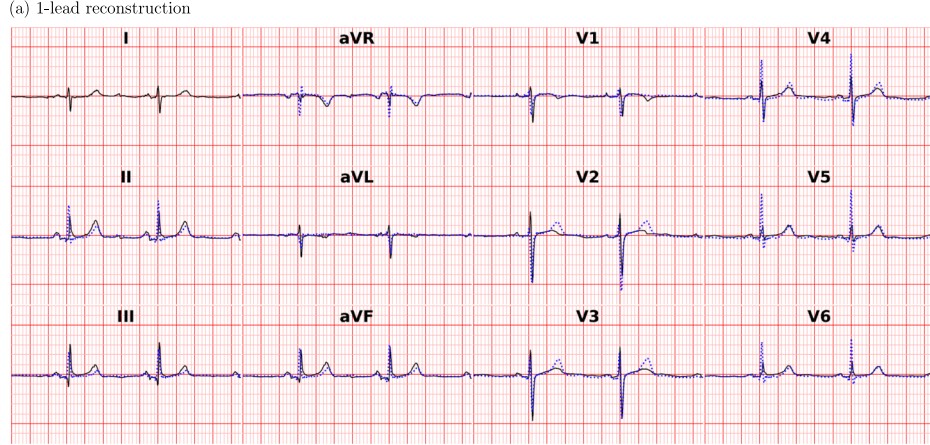

(a) 1-lead reconstruction

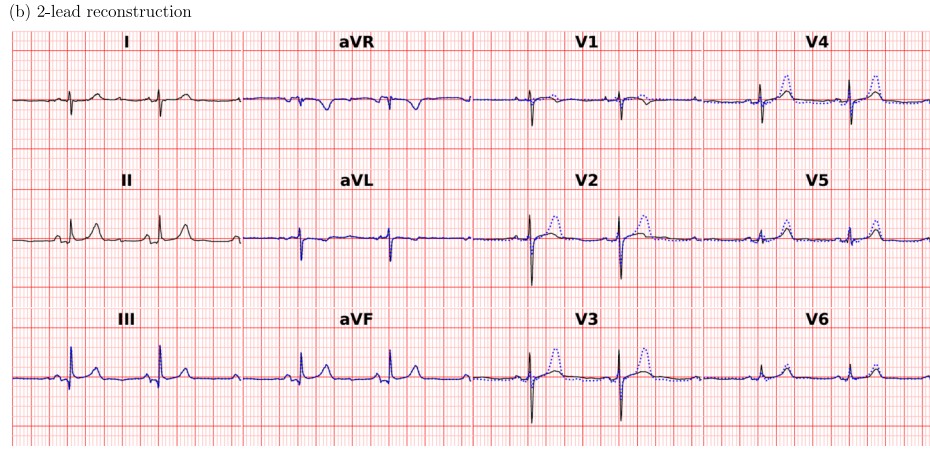

(b) 2-lead reconstruction

particularly important when comparing the performance of different models, as done here. A notable result from an F-test suggests that the variances are different, which implies that one model is more consistent than the other one in its predictions. This is an important consideration in our evaluation of the accuracy and reliability of ECG reconstructions. A $p$-value $< 0.05$ was considered statistically significant.

Reconstruction errors should be independent of the amplitude that is being reconstructed. Any evidence to the contrary could suggest a bias or regression towards the population mean, rather than accurately reflecting individual variations. Therefore, we plotted the difference in amplitude (reconstructed vs. real for R, S, STM, T) as a function of the real amplitude in a Bland-Altman-like plot. As the objective measurement of independence, we calculated Pearson's correlation coefficient between the difference and real amplitudes.

The reconstruction should not materially change the correlation between variables. If interlead correlations change materially, it is a sign that the model reconstructs leads based on one population mean and does not give personalized reconstructions based on the specific ECG. Therefore we calculated Pearson's correlation coefficient between markers (e.g., R peak) from the precordial leads (V1, V2, V3, V6) and lead I in the real ECGs and in the reconstructed ECGs, respectively. We also plotted the markers against one another to visually display any changes.

### Reporting summary

Further information on research design is available in the Nature Portfolio Reporting Summary linked to this article.

## Results

The GAN generated 12-lead ECGs with a natural appearance from both one- and two-lead ECGs. Figure 2 displays correlation plots depicting the

relationship between amplitudes in lead I (real input lead) and lead V3 (reconstructed lead), which is almost orthogonal to lead I. Similar figures for lead II instead of lead I are presented in Fig. S1 of the supplementary material. The figures show that real ECGs exhibit a poor correlation between R and T amplitudes in lead I and $V_n$, whereas the generated ECGs have a much higher correlation. This suggests that the GAN overuses input information in its reconstructions. Additionally, according to the Bland-Altman plots in Figs. 3 and 4, the GAN utilizing either lead I alone or both lead I and II to predict lead V3 (one of the orthogonal precordial leads) systematically overestimated low R- and T-wave amplitudes and underestimated large amplitudes not only in lead V3 but also in lead V6. Figure 5 shows an example of generated ECGs compared to the real ECG for both one and two lead reconstructions (more samples can be found in the Supplementary Figs. S2, S3, S4, S5, and S6).

Table 1 shows the 5, 10, 90, and 95% reconstruction error percentiles for all precordial leads. The results indicate that the reconstruction may erroneously contain clinically significant alterations that are not present in the real ECG and which may change the interpretation of the ECG. Table 2 displays correlation coefficients between the real and the reconstructed values, with the 2-lead reconstruction exhibiting higher values. The average root mean squared error (RMSE) for each lead is presented in Table 3 for both 1- and 2-lead reconstructions using the GAN, while the corresponding results for the UNet are presented in Table S1 in the supplementary. Table 4 exhibits the standard deviation and mean values for R, T, and S peaks and STM of real and generated data in leads V1, V2, V3, and V6.

## Discussion

Our study demonstrated that deep learning reconstruction from single- or dual-lead ECGs had substantial limitations, and was not clinically useful. The generated ECGs in the study had a normal appearance and by

**Table 1 | Percentile distribution of reconstruction error between real and reconstructed ECGs**

| ECG feature | Lead | 1-lead reconstruction | | | | 2-lead reconstruction | | | |
|---|---|---|---|---|---|---|---|---|---|
| | | 5% | 10% | 90% | 95% | 5% | 10% | 90% | 95% |
| RR interval (ms) | | 0 | 0 | 2 | 2 | 0 | 0 | 0 | 2 |
| QT interval (ms) | | −10 | −8 | 14 | 17 | −10 | −8 | 12 | 16 |
| PR interval (ms) | | −22 | −16 | 14 | 20 | −18 | −14 | 10 | 14 |
| QRS duration (ms) | | −16 | −12 | 10 | 14 | −14 | −10 | 10 | 14 |
| R amplitude (μV) | V1 | −229 | −175 | 92 | 107 | −230 | −161 | 68 | 88 |
| | V2 | −715 | −547 | 146 | 198 | −647 | −474 | 210 | 273 |
| | V3 | −953 | −717 | 362 | 479 | −923 | −658 | 435 | 542 |
| | V4 | −1125 | −854 | 542 | 686 | −1049 | −796 | 483 | 632 |
| | V5 | −823 | −630 | 416 | 547 | −788 | −576 | 303 | 393 |
| | V6 | −554 | −356 | 415 | 513 | −444 | −322 | 225 | 273 |
| T amplitude (μV) | V1 | −199 | −146 | 142 | 177 | −172 | −122 | 165 | 194 |
| | V2 | −429 | −337 | 209 | 269 | −332 | −240 | 303 | 361 |
| | V3 | −412 | −318 | 195 | 249 | −334 | −239 | 268 | 322 |
| | V4 | −347 | −268 | 152 | 200 | −264 | −210 | 185 | 229 |
| | V5 | −185 | −132 | 118 | 156 | −148 | −102 | 102 | 122 |
| | V6 | −122 | −83 | 98 | 122 | −88 | −59 | 68 | 88 |
| S amplitude (μV) | V1 | −547 | −443 | 365 | 587 | −478 | −366 | 303 | 507 |
| | V2 | −1074 | −805 | 405 | 538 | −959 | −713 | 425 | 578 |
| | V3 | −917 | −733 | 293 | 430 | −796 | −625 | 391 | 493 |
| | V4 | −757 | −591 | 185 | 249 | −610 | −415 | 278 | 348 |
| | V5 | −415 | −322 | 93 | 126 | −283 | −191 | 132 | 173 |
| | V6 | −170 | −126 | 39 | 53 | −87 | −59 | 68 | 85 |
| ST amplitude (μV) | V1 | −88 | −64 | 49 | 59 | −68 | −49 | 63 | 74 |
| | V2 | −230 | −161 | 78 | 98 | −171 | −117 | 117 | 142 |
| | V3 | −195 | −141 | 83 | 100 | −161 | −103 | 102 | 122 |
| | V4 | −136 | −93 | 63 | 78 | −117 | −78 | 63 | 78 |
| | V5 | −59 | −35 | 63 | 73 | −44 | −30 | 49 | 68 |
| | V6 | −49 | −34 | 43 | 49 | −35 | −25 | 39 | 49 |

Summary of the 5th, 10th, 90th, and 95th percentiles of the reconstruction error, calculated as the difference between the reconstructed ECG and the original ECG.

**Table 2 | $R^2$ correlation analysis of reconstructed ECGs across key intervals and amplitudes**

| ECG feature | Lead | 1-lead reconstruction | 2-lead reconstruction |
|---|---|---|---|
| RR interval | | 0.97 | 1.00 |
| QT interval | | 0.91 | 0.92 |
| PR interval | | 0.66 | 0.81 |
| QRS Duration | | 0.20 | 0.29 |
| R amplitude | V1 | 0.16 | 0.30 |
| | V2 | 0.07 | 0.10 |
| | V3 | 0.05 | 0.06 |
| | V6 | 0.09 | 0.47 |
| T amplitude | V1 | 0.09 | 0.11 |
| | V2 | 0.16 | 0.16 |
| | V3 | 0.20 | 0.25 |
| | V6 | 0.42 | 0.69 |
| S amplitude | V1 | 0.15 | 0.31 |
| | V2 | 0.02 | 0.09 |
| | V3 | 0.04 | 0.09 |
| | V6 | 0.16 | 0.55 |
| ST amplitude | V1 | 0.04 | 0.07 |
| | V2 | 0.36 | 0.40 |
| | V3 | 0.49 | 0.51 |
| | V6 | 0.50 | 0.65 |

$R^2$ correlation values between real and reconstructed data for different intervals, namely RR, QT, PR, and QRS and R, S, T, and ST amplitudes in leads V1, V2, V3, and V6. The results were derived from the GAN models trained using both 1-lead and 2-lead configurations.

themselves, could not be distinguished from real ECGs. However, a careful analysis of the mean amplitudes of different waves in the generated and real ECGs showed that they were significantly different. Importantly, nearly all leads and parameters exhibited less variation in the generated ECGs compared to the real ECGs, suggesting a regression towards the mean; and this phenomenon is clearly demonstrated in the Bland-Altman plots. Low amplitudes in the real ECG were increased in the synthetic ECGs and vice versa with high amplitudes in the real ECG, indicating that the network attempted to fit the mean of the population and not the individual ECG. The consequence would be that a patient with an acute myocardial infarction located orthogonal to the input lead would present with normal ST-amplitudes in the orthogonal leads generated from the population. We speculate that were the network instead trained on myocardial infarction patients, normal healthy subjects would exhibit signs of myocardial infarction in their orthogonal leads.

Several authors attempted to predict all 12 ECG leads from one. Lee et al. were the first to use GAN techniques to investigate 12-lead reconstruction from a single lead, a modified Lead II[5]. To generate the synthetic ECG, their proposed GAN used R-peak-aligned single heartbeats and data augmentation. They failed to present clinically relevant measures such as raw amplitudes in mV and instead used advanced mathematical algorithms, making it impossible to determine whether a clinically relevant difference existed. They also ignored the fact that the GAN reproduces the input population of ECGs nicely, but randomly and with no obvious relationship to the individual ECG. Guessing on the mean results in low amplitude mean absolute errors, but very skewed distributions. Only using Bland-Altman plots and outlier distributions can it be determined whether a low mean absolute error simply is due to guessing on the mean of the population in every case, as our study showed.

**Table 3 | Root mean squared error (RMSE) per lead comparing real and ECGs reconstructed with GAN**

| | Lead I | Lead II | V1 | V2 | V3 | V4 | V5 | V6 |
|---|---|---|---|---|---|---|---|---|
| 1-lead reconstruction (μV) | – | 79 | 76 | 136 | 136 | 120 | 96 | 79 |
| 2-lead reconstruction (μV) | – | – | 72 | 134 | 135 | 110 | 82 | 64 |
| SD real | 122 | 136 | 145 | 239 | 231 | 235 | 212 | 166 |

Average root mean squared error (RMSE) for every lead of the reconstructed ECGs. The results were derived from the GAN models trained using both 1-lead and 2-lead configurations.
*SD* standard deviation.

**Table 4 | Mean and standard deviation of ECG intervals and amplitudes across leads**

| ECG feature | Lead | Real data | 1-lead reconstruction | p(means) | p(variance) | 2-lead reconstruction | p(means) | p(variance) |
|---|---|---|---|---|---|---|---|---|
| RR interval (ms) | | 886 ± 161 | 887 ± 164 | 0.08 | 1.0 | 887 ± 162 | 1.0 | 0.4 |
| QT interval (ms) | | 397 ± 30 | 399 ± 31 | 2E−16 | 1.0 | 400 ± 30 | 2E−13 | 2E−13 |
| PR interval (ms) | | 160 ± 23 | 160 ± 20 | 0.3 | 0.001 | 159 ± 22 | 0.6 | 0.0001 |
| QRS duration (ms) | | 91 ± 9 | 90 ± 7 | 0.06 | 2E−16 | 91 ± 7 | 2E−12 | 0.2 |
| R amplitude (μV) | V1 | 161 ± 124 | 135 ± 51 | 2E−12 | 2E−16 | 126 ± 61 | 2E−16 | 2E−16 |
| | V2 | 504 ± 306 | 329 ± 87 | 2E−16 | 2E−16 | 401 ± 111 | 2E−16 | 2E−16 |
| | V3 | 856 ± 477 | 726 ± 132 | 2E−16 | 2E−16 | 784 ± 179 | 3E−6 | 2E−16 |
| | V6 | 1073 ± 325 | 1106 ± 192 | 0.001 | 2E−16 | 1038 ± 256 | 7E−6 | 2E−13 |
| T amplitude (μV) | V1 | −2 ± 121 | 0 ± 61 | 0.6 | 2E−16 | 26 ± 63 | 6E−14 | 2E−16 |
| | V2 | 419 ± 233 | 378 ± 112 | 7E−9 | 2E−16 | 466 ± 117 | 2E−11 | 2E−16 |
| | V3 | 481 ± 227 | 430 ± 123 | 4E−14 | 2E−16 | 514 ± 137 | 2E−7 | 2E−16 |
| | V6 | 236 ± 98 | 245 ± 76 | 0.0004 | 5E−14 | 242 ± 95 | 0.0008 | 0.4 |
| S amplitude (μV) | V1 | 820 ± 365 | 800 ± 164 | 0.08 | 2E−16 | 799 ± 215 | 0.04 | 2E−16 |
| | V2 | 1173 ± 491 | 1020 ± 154 | 2E−16 | 2E−16 | 1045 ± 213 | 2E−16 | 2E−16 |
| | V3 | 894 ± 427 | 703 ± 123 | 2E−16 | 2E−16 | 794 ± 182 | 2E−13 | 2E−16 |
| | V6 | 53 ± 79 | 23 ± 36 | 2E−16 | 2E−16 | 56 ± 71 | 0.03 | 0.001 |
| ST amplitude (μV) | V1 | 34 ± 29 | 29 ± 17 | 1E−7 | 2E−16 | 40 ± 24 | 9E−7 | 2E−10 |
| | V2 | 91 ± 66 | 70 ± 34 | 2E−16 | 2E−16 | 85 ± 39 | 0.0005 | 2E−16 |
| | V3 | 73 ± 66 | 72 ± 36 | 0.3 | 2E−16 | 75 ± 46 | 0.4 | 2E−16 |
| | V6 | 7 ± 27 | 9 ± 19 | 0.005 | 2E−16 | 12 ± 28 | 5E−9 | 0.1 |

Mean and standard deviation for RR, QT, PR intervals, QRS duration, and R, T, S peak amplitudes, and ST amplitudes of real and generated reconstructed data in leads V1, V2, V3, and V6. The results were derived from the GAN models trained using both 1-lead and 2-lead configurations. Real data vs. reconstructed data where tested for differences in mean using a t-test, and variance tested using an F-test.

Seo et al. used a patchGAN[5] to create 12-lead ECGs from 2.5 s ECG patches. They evaluated their network both with non-clinical methods using Fréchet distance but also used mean square error, interval, and amplitude measurements. According to their Bland-Altman plots, the 95 percent confidence interval for the difference between the real and synthetic GAN-generated ECG for RR intervals was approximately ±120 ms, QT intervals were approximately ±130 ms, and QRS intervals were approximately ±30 ms. These errors are far from clinically acceptable. The US Federal Drug Administration believes that a 10 ms prolongation of the QT interval has clinical significance, a 30 ms QRS prolongation can cause an incorrect diagnosis of a bundle branch block, and a 120 ms adaptation of the RR interval can change a heart rate of 100 to anything between 83 and 125.

A recent study[23] introduced a modified version of GAN, which incorporates two generators: one for expanding Lead I to generate the corresponding 12 leads and a second one that encodes and decodes Lead 1, acting like an autoencoder. The objective of the second generator is to approximate the latent vector of the main generator, ensuring that the 12-lead ECG retains essential characteristics of Lead I. The authors assessed their findings using a prediction model designed to detect left and right bundle branch blocks and atrial fibrillation, achieving favorable results. These features are not confined to one single lead but are more or less generalized to have information in many leads, therefore the good performance is not surprising. The dataset used for training the models is not publicly available.

Because we do not understand how advanced mathematical measures vary between healthy subjects and patients with heart disease, it becomes important to use relevant clinical outcomes as effect parameters.

We used one or two frontal limb leads (I or I+II) to train our GAN, as other authors[7] have done. In theory, these leads should provide information from one or two orthogonal directions rather than just one—either the X-axis or the frontal XY-plane. We would expect the dual-lead input to produce better results than the single-lead input, however, surprisingly, the network did not benefit when two limb leads were used as input instead of one. This is most likely because the X-axis and the XY-plane hold similarly little information on the Z-dimension, which dominates some of the precordial V leads. Thus, information from all three dimensions is needed in order to reconstruct an accurate, personalized 12-lead ECG.

Why is it then, that the GAN reconstruction clearly outperforms linear regression (Table S2). This is most likely due to the GAN's ability to extract vital biological information from the ECG in addition to the purely mathematical information, which the linear regression can also extract. It is well-known that sex, age, and body composition have large effects on the ECG, and neural networks may be able to extract these features from the ECG[24,25].

Patient characteristics—such as being young or old, female or male, and slim or overweight—are associated with increased amplitudes in the ECG in all dimensions and not just lead I. This biological correlation may explain why the neural networks are able to predict amplitudes in orthogonal leads, despite the fact that it should be mathematically impossible.

The study's limitations are that our network is suboptimal for solving the task of lead reconstruction. The accuracy of our model's predictions is inherently limited by the data on which it was trained, and a larger dataset might perform better. Future research could concentrate on improving these models to better capture this complexity, possibly by incorporating more patient-specific data into the model's design. Furthermore, experimenting with different machine learning techniques or lead configurations may yield more effective ECG reconstruction strategies.

In conclusion, the current study used deep learning techniques to investigate the reconstruction of 12-lead ECGs from single and dual leads and demonstrated that the reconstructions are not reliable for use in clinical practice. It seems more clinically useful to improve methods to gather more information from single or reduced leads using AI instead of solving a mathematically impossible problem. However, we can investigate different hyper-parameters, such as learning rates, activation functions, and different numbers of layers, to get improved results in future studies, as hyper-parameter tuning was identified as one of the limitations of this study.

## Data availability

The ECG data supporting the findings of this study are publicly available at Physionet.org with the identifier: https://doi.org/10.13026/kfzx-aw45[12]. The data used to generate the plots in Figs. 2, 3, 4 is available in our GitHub repository[26] as a CSV file (*12SL-ecg.csv*), along with the corresponding code for plot generation.

## Code availability

The code repository[26] contains the source code for the showcased networks, including links to checkpoints of pre-trained models for reconstruction purposes, along with a collection of generated samples.

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

## Acknowledgements

The research presented in this paper has benefited from the Experimental Infrastructure for Exploration of Exascale Computing (eX3), which is financially supported by the Research Council of Norway under contract 270053.

## Author contributions

O.P., A.D., A.R.S., J.K.K., and V.T. conceived the experiment(s). O.P., A.D., and T.W. conducted the experiment(s). O.P., A.D., J.L.I., C.G., M.A.R., A.R.S., J.K.K., and V.T. analyzed the results. All authors reviewed and revised the manuscript.

## Funding

## Competing interests

Arun Sridhar was previously an Editorial Board Member for Communications Medicine, and is currently a Guest Editor, but was not involved in the editorial review or peer review, nor the decision to publish this article. All other authors declare no competing interests.
