## [Transparent Peer Review file · Communications Medicine]

Evaluating the Feasibility of 12-Lead ECG Reconstruction from Limited Leads Using Deep Learning

Corresponding Author: Dr Vajira Thambawita

Version 0:

Reviewer comments:

Reviewer #1

(Remarks to the Author)

In their article, Presacan and coworkers evaluate whether a 12 lead ECG could be reconstructed from a one or two lead ECG using deep neural networks. The authors use data from the PTB-XL dataset. The main finding of the study was that this is not possible.

The manuscript is interesting and well performed. I also believe that the work is important, especially in view of the current hype in the field of artificial intelligence. I have following comments:

Comments:

- 1) The study tests the hypothesis that electric biological information derived from multiple sensors (i.e. 12 leads) can be predicted from much less sensors (i.e. one or two leads). That fact that this is not possible (which can be expected) does not mean that the medical / clinical information embedded in a 12 leads ECG cannot be derived from much less leads. In other words: Reconstructed T waves may look differently, but a myocardial infarction could be recognized.
- 2) „From a mathematical perspective, only three orthogonal leads should be required to characterize the electrical vector in space; a concept well-known from the 3-lead Frank vectorcardiogram“ I am not sure whether this statement is correct. Maybe the authors could have a look at the concept of QRS microfragmentation (European Heart Journal, Volume 43, Issue 40, 21 October 2022, Pages 4177–4191)
- 3) “However, the lack of dimensions may result in substantial loss of spatial information, which is essential for diagnosing myocardial infarction, hypertrophy, and other localized cardiac disorders.“ You may add repolarization abnormalities which -from a clinical standpoint- are most relevant
- 4) The title suggests much more than it is. This paper develops an algorithm based on GAN but does not compare it with other state-of-the-art GANs for this application.
- 5) I am surprised that lead I vs. lead I + II does not make a big difference. If you have access to I + II, you should be able to derive all limb derivatives. You can also see in Fig.4(b) that these derivatives are generated much better than in the case of Fig.4(a).
- 6) A recent and well-published paper is completely ignored. This should be used as a benchmark. If the paper was submitted recently (the manuscript says June 2023), this paper cannot be ignored: https://doi.org/10.1007/978-3-031-43990-2_18

Reviewer #2

(Remarks to the Author)

In this paper, authors try to use deep learning for 1-to-12 ECG reconstruction. It is known that the generated signals could exhibit a normal appearance, but there are some deficiencies in this research field. These results indicate the reconstructed ECG has reduced ability, that is a regress to the population mean. However, some issues should be considered and discussed in this study, and some suggestions could improve the manuscript quality, as follows.

1. The manuscript structure could be improved. Is it suitable to present the 1-lead reconstruction results and 2-lead reconstruction results at the same time? The manuscript title is about one-to-12-lead ECG reconstruction.
2. Authors show some experimental results to prove its disadvantages for deep learning to reconstruct 12-lead ECG with limited-lead ECG, as the difference between the generated and real ECG signals. However, it might be influenced by the designed model, like the output activation of the generator is tanh. Could you adopt some more effective activation or linear layer? A well-designed model may be all you need.
3. Authors might conduct these experiments on the previous studies and models, and use the models of Lee et al (IEEE Journal of Biomedical and Health Informatics 24, 1265–1275) and Seo et al (Computer Methods and Programs in Biomedicine 221, 106858).

4. The experimental results in figure 4 seem good, can you supplement some detailed metric to evaluate the generate signal directly, like Mean square error or R2.
5. Why the input shape of discriminator is 8*800, not 8*5000?
6. If the 1-to-12-lead ECG reconstruction task is difficult for deep learning, authors could show their suggestions for the future work in this research field.

Version 1:

Reviewer comments:

Reviewer #1

(Remarks to the Author)

The authors adequately responded to the comments of both reviewers. I have no further issues.

Reviewer #2

(Remarks to the Author)

None
